# Estimating the Physicochemical and Antioxidant Properties of Hardy Kiwi (*Actinidia arguta*) Treated with 1-Methylcyclopropene during Storage

Tomasz Krupa *, Andrii Kistechok and Kazimierz Tomala

Department of Pomology and Horticulture Economics, Institute of Horticultural Sciences, Warsaw University of Life Sciences (SGGW-WULS), 159C Nowoursynowska Street, 02-787 Warsaw, Poland
* Correspondence: tomasz_krupa@sggw.edu.pl; Tel.: +48-22-593-21-04

**Abstract:** In fruit storage, new methods are being sought to extend the distribution period while maintaining the highest quality parameters of the fruit, i.e., the physical and chemical characteristics of the fruit, but also the health-promoting properties. One method is to treat the fruit with 1-MCP, which effectively inhibits fruit ripening, since the main reasons for limiting the distribution of minikiwi fruit are rapid ripening and the loss of firmness. It is also highlighted that minikiwi is a source of antioxidants, which, as highly reactive compounds, are quickly degraded during storage. This study evaluates the effectiveness of using 1-MCP to reduce minikiwi softening and maintain the high antioxidant properties of the fruit. In the experiment, minikiwi fruits of the 'Ananasnaya' cultivar were used. After harvesting, the fruits were treated with 1-methylcyclopropene at a concentration of 0.65 μL/L. Fruits treated in this way were stored in ordinary cold storage (NA) and under low oxygen concentration (ULO) conditions for a period of 12 weeks. The fruits' physicochemical properties, sugar and acid contents and antioxidant potential, which consisted of ascorbic acid, polyphenols, phenolic acids, flavonols and flavan-3-ols, were evaluated. The application of 1-MCP is effective in reducing the loss of firmness in the minikiwi during storage, but the blocking of minikiwi ripening by 1-MCP makes the fruit less sweet and more acidic. Fruits treated with 1-MCP had higher antioxidant activity and a higher content of biologically active compounds. The effect of 1-MCP is stronger for flavan-3-ols, but slightly weaker for phenolic acids. Fruits treated with 1-MCP have a higher antioxidant potential than untreated fruit after a long period of storage.

**Keywords:** hardy kiwi; 1-MCP; fruit quality; storage; antioxidant properties

## 1. Introduction

The climate warming observed in the past decade has favoured the cultivation of 'warm-loving' species that were previously impossible to cultivate in colder regions of the world. An example is the minikiwi (*Actinidia arguta* (Sieb. & Zucc.) Planch. ex Miq.) species, which has been winning not only the hearts of growers, but also the tables of consumers for more than 20 years [1,2]. Current consumer preferences focus on fruits that are distinctive in appearance, taste and nutritional properties [3,4]. Minikiwi, unlike the popular kiwifruit, can be eaten with the peel, which enhances the nutritional value of this fruit because, as in other species, the peel of the fruit contains more health-promoting compounds than the pulp itself [5–7]. The nutraceutical content of the fruit is mainly dependent on the genetic factor [8,9]. However, environmental conditions, i.e., the cultivation location, the climatic conditions or the agrotechnical or soil treatments, modify the contents of biologically active compounds [10–14]. A strong determinant of the intrinsic quality of the fruit is the maturity stage, which is determined by the ripeness of the fruit at harvest [15–18] or the conditions under which the fruit ripens [19,20]. The harvest date and new technologies can improve the storability of fresh strawberry fruit [21].

The high nutraceutical content of minikiwi is causing it to win more and more fans worldwide. Cultivars with larger sizes, but also with higher nutrient contents or storability, are being sought [22]. It is the short distribution period of the fruit that is still the main problem in plantation development and fruit distribution [19,23,24]. Therefore, the attention of scientific centres is focused on the search for new technological solutions that can improve the storability of fresh fruit [20,25,26], as well as their use as a source of biologically active compounds in the production of food additives or cosmetics [7,27,28].

A commonly used inhibitor of the ripening processes in apples is 1-methylcyclopropene (1-MCP). Hoang et al. [29] and Kolniak-Ostek et al. [30] claim that 1-MCP treatment delays the loss of phenolic compounds in apples during storage. Also, according to MacLean et al. [31] the application of 1-MCP prevents anthocyanin degradation in apples, but does not affect changes in flavonols and flavan-3-ols. 1-MCP reduces the loss of antioxidant potential in pears and, in combination with controlled atmosphere conditions, may even increase the antioxidant potential [32]. The use of inhibitors, i.e., aminoethoxyvinylglycine (AVG) or 1-methylcyclopropene (1-MCP), effectively blocks ageing or oxidation processes [33,34]. The loss of general antioxidant capacity begins when the integrity of the cell associated with mechanical damage or the natural ageing process is broken. The initiation of catalysis of the metabolism and the degradation of phenolic compounds via enzymes such as esterases, glycosidases and decarboxylases result in a significant loss of the fruit's health-promoting properties. 1-Methylcyclopropene (1-MCP) blocks the binding of ethylene to its receptor, which results in a slowdown in the ethylene production and inhibits the ripening processes that determine this phytohormone [35]. The efficacy of using 1-MCP treatment to improve storability has been widely demonstrated for 'Hayward' kiwifruit [36–38], but despite the promising results, few studies have evaluated the effect of 1-MCP treatment on minikiwi [39–41]. Still unexplained is the blocking of aroma synthesis and the deterioration of flavour in fruit treated with 1-MCP. Previous studies have shown the high effectiveness of 1-MCP in reducing the softening of fruit, even when harvested at a more advanced stage of ripeness [41]. The issue of the effect of 1-MCP on the intrinsic quality traits determined by a group of bioactive compounds needs clarification.

The aim of this study is to evaluate the effect of the post-harvest application of 1-methylcyclopropene on the intrinsic quality of minikiwi fruit during storage under conventional versus controlled-atmosphere cold storage conditions. In the study, the fruit quality is described by physicochemical characteristics as well as changes in the contents of the biologically active compounds that are valuable to human health.

## 2. Materials and Methods

### 2.1. Outline of the Experiment

The fruit originated from the experimental field of the Department of Pomology and Horticultural Economics, Warsaw University of Life Sciences (WULS-SGGW), located in Warsaw, central Poland (52.259° N, 21.020° E). The fruits used in the experiment were *Actinidia arguta* ((Sieb. & Zucc.) Planch. ex Miq.), commonly known as minikiwi (in Poland), hardy kiwi, baby kiwi, etc. The 'Ananasnaya' cultivar was used to evaluate the physical and chemical properties of the fruit, which is the basic cultivar grown in the United States and is the most widely grown minikiwi fruit in the world [42]. Minikiwi plants were grown in a T shape with a spacing of 4 m × 4 m. The plants were planted in 2009, and managed according to good agricultural practice. The fruit was harvested on September 24 in 2015 (the first year of the study) and September 17 in 2017 (the second year of the study). Fruits were hand-harvested into well-ventilated PVC containers (500 g) (Best Opakowania, Pniewy, Poland) when the fruits reached harvest maturity (6–7 °Brix) based on guidelines from the literature and our own previous research [43,44]. Immediately after harvesting, the fruit packages were transported to the cold store of the WULS-SGGW, Institute of Horticultural Sciences, where they were placed under refrigeration at 1 °C, Rh = 75–80%, to cool the berries. After approximately 24 h of refrigeration, half of the fruits were treated with 1-MCP (1-methylcyclopropene) (SmartFresh ProTabs™, AgroFresh Solutions Inc., Philadelphia,

PA, USA). A 24 h treatment of the fruit with 1-MCP at a concentration of 0.65 μL/L under refrigerated conditions (1 °C, Rh = 75–80%) was applied as recommended in the literature [25]. Fruits that were not treated with 1-MCP were also left under refrigerated conditions for 24 h. The two fruit batches were then divided into two groups (separately for 1-MCP-treated and untreated fruits) and placed in two experimental containers (1 m$^3$ in area) equipped with an Oxystat 200 automatic gas control system (David Bishop Ltd., Heathfield, UK), providing monitoring of $CO_2$ and $O_2$ contents every 4 h. The following refrigerated conditions were maintained in the experimental containers:

1. Suitable for conventional cold storage (0.1% $CO_2$:21% $O_2$)—NA;
2. Ultra-low oxygen (1.5% $CO_2$:1.5% $O_2$)—ULO.

In the experimental containers, the temperature was maintained at 1 °C (±0.5 °C), and the relative humidity was approximately 95% (±3%). In the first season of the study, the fruit was stored for 8 weeks as recommended in the literature [19,25]. In the second season of the study, the storage period was extended to 12 weeks due to the positive effects of 1-MCP application. Fruit samples were taken for physicochemical analyses every 2 weeks in both storage seasons. Laboratory analyses were repeated three times, using 20–30 fruits for each test.

### 2.2. Chemicals and Reagents

All reagents used for HPLC were of HPLC grade and purchased from Sigma-Aldrich (Poznań, Poland) and Merck (Warsaw, Poland). Other chemicals were of analytical purity grade and purchased from Alchem (Warsaw, Poland).

### 2.3. Analytical Methods

Physicochemical indicators (firmness, extract content and acidity) describing the parameters most relevant to consumer preferences and indicators describing the nutritional and biological active compounds of the fruit (sugars and acids, vitamin C, polyphenols, flavonoids, phenolic acids, etc.) were used to assess the fruit. Analyses were carried out immediately after fruit harvest and at specific dates after storage.

Fruit firmness (FF), according to a previously described method [16], was determined as the value of the force required to penetrate the fruit with a 4.5 mm diameter probe. FF was determined using an Instron 5542 penetrometer (Instron, High Wycombe, UK) on 20 fruits. Each fruit was subjected to compression twice (on opposite sides). The probe speed was 240 mm$^{-1}$, and the fruits were penetrated to a depth of 5 mm. FF is expressed in Newton (N).

The soluble solids content (SSC) was determined refractometrically, in accordance with Polish Standard PN-EN 12,143:2000 [45] (developed by the Polish Committee for Standardisation) according to the previously described method [44]. Determinations were performed in juice squeezed from 20 fruits. SSC was measured on an Atago Palette PR-32 alpha digital refractometer (Atago, Tokyo, Japan) at 20 °C. The results are expressed in °Brix. Similarly, in the juice obtained from the previously used 20 fruits, the titratable acidity (TA) was determined according to Polish Standard PN-EN 12,147:2000 [46]. Acidity was assessed via titration with 0.1 N sodium hydroxide (NaOH) of a 10-fold diluted aqueous extract of the fruit juice (1:10, *v:v*) to an end point of pH 8.1. A TitroLine 5000 system (Si Analytics, Mainz, Germany) was used for the analysis. Results are expressed as a percentage of anhydrous citric acid.

Sugars and organic acids were determined via HPLC-RI, as described previously by Zielinski et al. [47], and are expressed as grams of total sugar content or organic acid per 100 g F.W. We determined the antioxidant activity according to the method of Saint Criq de Gaulejac et al. [48], which is based on the reduction in free radicals derived from DPPH$^+$ (1,1-diphenyl-2-picrylhydrazine, Sigma-Aldrich, Poznań, Poland). The antioxidant activity was calculated from absorbance measurements for the corresponding sample (fruit extract + DPPH$^+$) taken after 20 min at λ = 517 nm relative to the control sample ($H_2O$). Results are expressed as mg ascorbic acid equivalent per g$^{-1}$ FW (fresh weight). Determina-

tion of ascorbic acid content was performed according to the method described in an earlier study [20]. Identification and quantitative analysis of ascorbic acid were performed with a 200-series HPLC system (Perkin Elmer Ing, Waltham, MA, USA) equipped with a diode array detector (UV-DAD), using a Spheri-5 RP-18 column (5 μm, 220 mm × 4.6 mm, Brownlee Columns, Waltham, MA, USA) at a flow rate of 1.0 mL/min and detection at 245 nm. Total phenolic content (TPC) was determined spectrophotometrically [49] using the Folin–Ciocalteu reagent. A Marcel 330S PRO spectrophotometer (Marcel, Zielonka, Poland) was used to measure the absorbance of the solution, and the measurement was performed at λ = 700 nm. The results presented were converted to gallic acid. Separation of phenolic compounds was performed using the HPLC technique described in our previous study [20]. The separation and contents of phenolic compounds were analysed using a Perkin-Elmer 200 series HPLC kit with a diode array detector (DAD). Separation was performed using a LiChroCART 125-3 column (Merck KGaA, Darmstadt, Germany) at a flow rate of 1 mL/min. Phenolic compounds were detected at 254, 280, 320, and 360 nm wavelengths by comparing retention times on achieved chromatograms with those of standards. The total phenolic acids consisted of identified derivatives of chlorogenic, neochlorogenic, caffeic, *p*-coumaric and other derivative acids. In the flavonols group, quercetin derivatives (quercetin-galactoside, quercetin-rutinoside, quercetin-glucoside, etc.) and keampferol derivatives (keampferol-rutinoside and keampferol-galactoside) were identified. The sum of compounds in the flavan-3-ol group was (+)-catechin and L-epicatechin.

### 2.4. Statistical Analysis

The results were analysed statistically in the Statistica 12.5 program (StatSoft Polska, Krakow, Poland). The effect of experimental variables, i.e., the post-harvest treatment of fruit with 1-MCP (+/−1-MCP), storage conditions (NA, ULO), and storage duration, on quality parameters of minikiwi were analysed using multifactor analysis of variance (ANOVA). A Newman–Keuls test was used to evaluate the significance of differences between the means, accepting the significance level as 5% or 1%. Mean and standard deviation were also reported for all measured parameters.

## 3. Results and Discussion

### 3.1. Characteristics of the Fruit after Harvesting

The characteristics of the physicochemical indices and the values of the bioactive compounds immediately after fruit harvest are shown in Table 1.

**Table 1.** Characteristics of internal and external fruit quality indicators after harvesting.

| Characteristics | 2015 | 2017 |
|---|---|---|
| Firmness (N) | $50.2 \pm 2.4$ | $52.4 \pm 2.6$ |
| Soluble solids content (°Brix) | $7.0 \pm 0.3$ | $6.6 \pm 0.2$ |
| Acidity (% citric acid) | $0.98 \pm 0.05$ | $1.08 \pm 0.07$ |
| Glucose (mg·100 g$^{-1}$ FW) | $1.86 \pm 0.04$ | $1.73 \pm 0.04$ |
| Fructose (mg·100 g$^{-1}$ FW) | $2.18 \pm 0.07$ | $2.25 \pm 0.02$ |
| Sucrose (mg·100 g$^{-1}$ FW) | $6.70 \pm 0.03$ | $6.34 \pm 0.13$ |
| Citric acid (mg·100 g$^{-1}$ FW) | $0.69 \pm 0.01$ | $0.88 \pm 0.02$ |
| Malic acid (mg·100 g$^{-1}$ FW) | $0.167 \pm 0.002$ | $0.129 \pm 0.05$ |
| Ascorbic acid (mg·100 g g$^{-1}$ FW) | $85.2 \pm 6.4$ | $76.6 \pm 7.1$ |
| TPC (mg·100 g$^{-1}$ FW) | $101.5 \pm 8.1$ | $106.0 \pm 7.6$ |
| Phenolic acids (mg·100 g$^{-1}$ FW) | $1.71 \pm 0.11$ | $2.04 \pm 0.09$ |
| Flavonols (mg·100 g$^{-1}$ FW) | $5.41 \pm 0.62$ | $4.12 \pm 0.29$ |
| Flavan-3-ol (mg·100 g$^{-1}$ FW) | $0.62 \pm 0.02$ | $0.52 \pm 0.02$ |
| AA (mg EAA g$^{-1}$ FW) | $0.98 \pm 0.11$ | $0.82 \pm 0.03$ |

AA, antioxidant activity; EAA, ascorbic acid equivalent; TPC, total polyphenol content; $\pm$, standard deviation; N, Newton; FW, fresh weight.

The fruit harvested in the first season was characterised by a slightly more advanced degree of post-harvest ripeness than the fruit from the second season. This was clearly evidenced by the physicochemical characteristics of the fruit in 2015, i.e., slightly lower firmness and acidity but a higher soluble solids content (SSC). The glucose and fructose contents of the fruit harvested in both years were at fairly similar levels, but the sucrose content was higher in the post-harvest fruit in 2015. Similar relationships apply to the acidity and individual acid contents. The titratable acidity as well as the citric acid content in the fruit from the first year of the study were lower than in the second year. Unexpectedly, the malic acid content was characterised inversely. In 2015, the post-harvest fruit was characterised by a higher ascorbic acid content and higher antioxidant activity. The levels of individual groups of phenolic compounds were variable depending on the year of the study. The total polyphenol content and phenolic acid content were distinguished by lower values in the first year than in the second year of the study. In contrast, the contents of flavonol and flavan-3-ol derivatives were higher in the fruit from the first season than from the second season. The different levels of fruit quality are related to a number of factors affecting the physical characteristics of the fruits as well as their antioxidant profiles. The occurrence of year-to-year variations is becoming the rule in long-term studies, especially in years with numerous weather 'anomalies' associated with climate change. Many reports point out that annual weather variability very strongly determines the quality and nutritional characteristics of the fruits of different species [50–53]. In-season weather conditions can significantly alter the quantitative and/or qualitative characteristics of internal fruit quality [50]. The external and internal qualities of fruit are made up of many variables (weather components and soil conditions); therefore, it is difficult to estimate whether the level of an individual indicator was influenced by one or many factors. The physiological state of the fruit, understood as the degree of ripeness, is an important factor influencing fruit characteristics. In more mature fruit, the firmness and acidity will be distinguished by lower values, and the soluble matter content will be distinguished by a higher value [22,36,41]. Similar changes occurring during fruit ripening will affect biologically active compounds [20,22,39].

### 3.2. Physical and Chemical Properties

The softening of minikiwi is an important cause of fruit storage and distribution problems [54]. One of the causes of fruit softening is the activation of the enzymes responsible for the softening process, i.e., polygalacturonase, esterase, pectin and endo-1,4β-D-glucanase, by ethylene [33]. The inactivation of the enzymes by 1-methylcyclopropene makes it possible, on the one hand, to inhibit fruit softening processes, and on the other hand, to slow down the synthesis of ethylene from ACC (1-aminocyclopropane-1-carboxylic acid) [37,38,55]. The effect of post-harvest treatment with 1-MCP on the fruit firmness of minikiwi is shown in Tables 2 and S1. The use of 1-methylcyclopropene strongly inhibited fruit softening regardless of the conditions under which the fruits were stored. Beneficial effects of 1-MCP were recorded throughout the entire storage period of the minikiwi. The fruits treated with 1-MCP were characterised by firmness that was 2–4 times higher than the untreated fruit, and the differences increased with the increasing storage period. The inhibition of ethylene synthesis is possible under refrigerated conditions by lowering the air temperature [15], reducing the oxygen concentration or increasing the carbon dioxide levels [20]. Our research as well as reports in the literature [2,16,19,54] indicate that low-temperature storage (0–1 °C) is not sufficiently effective in maintaining the high quality of minikiwi, and new technologies are still being sought to efficiently inhibit respiration processes in the fruit and the loss of quality during the post-harvest life of the fruit. The methods evaluated so far, such as the post-harvest treatment of minikiwi [56], modified atmosphere packaging technology [26,57], the use of ozone [24] or dynamically controlled atmosphere technology [20], have proven to be insufficiently effective. In our study, the interaction between the use of 1-MCP and the storage conditions was demonstrated, illustrated by the synergistic effect of ULO conditions and 1-MCP treatment (Table S1). Similarly,

the synergistic interaction of all the experimental factors was proven; the highest firmness throughout the experiment was characterised by the fruit treated with 1-MCP and stored under ULO conditions. Synergistic effects of two and more factors limiting the softening of kiwifruit and other species were observed in studies on 1-MCP in recent years [29,38,41,58] The binding of 1-MCP (ethylene homologue) to the ethylene receptors of enzymes results in the prolonged storage of products with almost unchanged quality parameters in different fruit species [35,58,59].

**Table 2.** Estimation of the effects of 1-MCP treatment and storage technology on the firmness (N) of minikiwi after storage.

| Storage Technology | Post-Harvest Treatment | Time of Storage (Weeks) | | | | | | p-Value |
|---|---|---|---|---|---|---|---|---|
| | | 2 | 4 | 6 | 8 | 10 | 12 | |
| | | 2015 | | | | | | |
| NA | +1-MCP | 27.6 ± 4.8 | 18.5 ± 1.4 | 11.5 ± 1.2 | 7.1 ± 0.2 | - | - | <0.001 |
| | −1-MCP | 15.7 ± 1.2 | 9.5 ± 0.7 | 5.9 ± 0.6 | 4.2 ± 0.2 | | | <0.001 |
| | p-value | 0.026 | 0.001 | 0.003 | <0.001 | | | |
| ULO | +1-MCP | 37.3 ± 0.9 | 32.7 ± 0.2 | 27.1 ± 1.9 | 23.8 ± 1.0 | - | - | <0.001 |
| | −1-MCP | 28.5 ± 2.0 | 15.7 ± 1.4 | 8.8 ± 1.4 | 5.3 ± 0.7 | - | - | <0.001 |
| | p-value | 0.004 | <0.001 | <0.001 | <0.001 | | | |
| | | 2017 | | | | | | |
| NA | +1-MCP | 32.5 ± 3.8 | 22.4 ± 1.5 | 17.1 ± 0.6 | 11.5 ± 1.3 | 7.5 ± 1.4 | 6.8 ± 0.6 | <0.001 |
| | −1-MCP | 21.9 ± 0.9 | 12.9 ± 0.3 | 7.5 ± 0.4 | 3.4 ± 0.5 | 2.4 ± 0.1 | 1.5 ± 0.2 | <0.001 |
| | p-value | 0.0185 | <0.001 | <0.001 | 0.001 | 0.006 | <0.001 | |
| ULO | +1-MCP | 40.6 ± 0.9 | 32.8 ± 0.9 | 29.6 ± 1.5 | 24.8 ± 1.1 | 18.1 ± 1.1 | 14.2 ± 1.3 | <0.001 |
| | −1-MCP | 30.7 ± 2.0 | 15.2 ± 2.1 | 9.0 ± 0.2 | 5.4 ± 1.0 | 2.9 ± 0.1 | 2.9 ± 0.6 | <0.001 |
| | p-value | 0.002 | <0.001 | <0.001 | <0.001 | <0.001 | <0.001 | |

NA, conventional cold storage, 0.1% $CO_2$:21.0% $O_2$; ULO, ultra-low oxygen, 1.5% $CO_2$:1.5% $O_2$; +1-MCP, with 1-methylcyclopropene; −1-MCP, without 1-methylcyclopropene; N, Newton; ±, standard deviation.

Among the instrumental distinguishing features of fruit quality are often their internal characteristics related to sweetness and acidity, i.e., the soluble solids content and titratable acidity. The ratio between these indicators is particularly emphasised. There is a lack of information in the literature on the effect of post-harvest treatment with 1-MCP on the soluble solids content or acidity of minikiwi fruit. Based on studies on the use of 1-MCP in the storage of other species, it can be shown that the effect of 1-MCP is not clear-cut and depends on the year of study and the species, and that the soluble solids content increases in the fruit regardless of the application of 1-MCP [30,36,41]. In our study, we found that the effect of 1-MCP on the SSC in fruit depended on the year of study (Tables 3 and S1). The effect of the ripening inhibitor was proven only in the second storage season; however, the 1-MCP-treated fruit were characterised by a lower SSC in both seasons. When considering the SSC at individual storage dates, it was found that only a few dates showed no significant effect of 1-MCP, and often, the effect was at a very high level of significance (Table 3). In our earlier studies, the effect of 1-MCP on the soluble solids content of fruit stored under conventional cold storage (NA) conditions was reported, but it was only reported in a few cases of storage under ULO [41]. An analysis of the results showed an interaction between the use of 1-MCP and the other experimental factors (Table S1). The fruits that were treated post harvest and stored in ULO were characterised by a lower SSC, irrespective of the date of analysis. A steady increase in the soluble solids content was recorded throughout the experiment, and it was slower in the fruit treated with 1-MCP. Unexpectedly, extending the storage period from 8 to 12 weeks revealed a change in the

upward trend of the SSC. Namely, a decrease in the soluble solids content was found after 12 weeks of storage for the fruit that was not treated with 1-MCP, regardless of the storage conditions. A similar relationship was not observed for the fruit treated with 1-MCP. It should be noted that the fruit treated with 1-MCP, regardless of the length of the storage period, did not obtain such a high SSC value as the untreated fruit, with the exception of the fruit stored in NA in 2015. An increase in the SSC during the storage period is observed in most species. Many authors evaluating the effect of 1-MCP emphasise that fruits treated with this inhibitor often do not achieve as high SSCs as untreated fruits [25,30,59]. This is particularly true for climacteric fruits such as kiwi, apple, pear, etc. It is extremely rare to observe a decrease in the SSC towards the end of the storage period, which is related to the consumption of sugars during respiration, and which can already be interpreted as the beginning of the fruit's over-ripening process. In the case of kiwifruit or minikiwi, the loss of monosaccharides may occur more rapidly, as these fruits do not contain significant amounts of starch, a polysaccharide whose breakdown via hydrolysis leads to an increase in the contents of monosaccharides and sucrose in the plants [60,61]. The second indicator that is responsible for fruit palatability is acidity. In our study, we showed a highly significant effect of 1-MCP on the acidity of minikiwi in both years of the study; however, this effect was only observed for the minikiwi stored in NA (Tables 4 and S1). The storage conditions were not shown to have an effect in the experiment. Many authors emphasise that the use of 1-MCP determines the rate of fruit acidity loss, in favour of the use of 1-MCP—such fruits have a higher acidity [34,62]. It is interesting to note that the effect of the storage period was only proven for the fruit that was not treated with 1-MCP, regardless of the storage conditions. Comparing the results obtained from both years of the study with those of the previously described studies [41,54], it can be concluded that the effect of 1-MCP on the acidity stabilisation of minikiwi depends on a number of additional factors, i.e., the year of study, the storage conditions and the fruit maturity status.

**Table 3.** Estimation of the effects of 1-MCP treatment and storage technology on the soluble solids content (°Brix) of minikiwi after storage.

| Storage Technology | Post-Harvest Treatment | Time of Storage (Weeks) | | | | | | *p*-Value |
|---|---|---|---|---|---|---|---|---|
| | | **2** | **4** | **6** | **8** | **10** | **12** | |
| | | **2015** | | | | | | |
| NA | +1-MCP | $10.3 \pm 0.3$ | $13.1 \pm 0.2$ | $14.8 \pm 0.1$ | $15.3 \pm 0.2$ | - | - | <0.001 |
| | −1-MCP | $12.0 \pm 0.2$ | $14.5 \pm 0.1$ | $15.6 \pm 0.1$ | $15.2 \pm 0.2$ | | | <0.001 |
| | *p*-value | 0.002 | <0.001 | <0.001 | 0.872 | | | |
| ULO | +1-MCP | $9.7 \pm 0.4$ | $10.7 \pm 0.2$ | $12.4 \pm 0.4$ | $13.3 \pm 0.4$ | - | - | <0.001 |
| | −1-MCP | $9.8 \pm 0.7$ | $12.5 \pm 0.3$ | $14.1 \pm 0.2$ | $14.4 \pm 0.4$ | - | - | <0.001 |
| | *p*-value | 0.857 | 0.002 | 0.007 | 0.050 | | | |
| | | **2017** | | | | | | |
| NA | +1-MCP | $11.6 \pm 0.5$ | $13.3 \pm 0.4$ | $14.6 \pm 0.3$ | $14.9 \pm 0.5$ | $15.9 \pm 0.2$ | $15.8 \pm 0.3$ | <0.001 |
| | −1-MCP | $11.8 \pm 0.5$ | $14.9 \pm 0.6$ | $17.2 \pm 0.4$ | $17.4 \pm 0.2$ | $16.5 \pm 0.1$ | $14.7 \pm 0.4$ | <0.001 |
| | *p*-value | 0.718 | 0.034 | <0.001 | <0.001 | 0.026 | 0.040 | |
| ULO | +1-MCP | $9.7 \pm 0.5$ | $11.6 \pm 0.5$ | $12.4 \pm 0.2$ | $13.9 \pm 0.3$ | $14.0 \pm 0.2$ | $14.1 \pm 0.3$ | <0.001 |
| | −1-MCP | $11.0 \pm 0.2$ | $13.2 \pm 0.9$ | $14.8 \pm 0.5$ | $16.6 \pm 0.3$ | $16.0 \pm 0.1$ | $15.4 \pm 0.3$ | <0.001 |
| | *p*-value | 0.031 | 0.092 | 0.003 | <0.001 | <0.001 | 0.013 | |

NA, conventional cold storage, 0.1% $CO_2$:21.0% $O_2$; ULO, ultra-low oxygen, 1.5% $CO_2$:1.5% $O_2$; +1-MCP, with 1-methylcyclopropene; −1-MCP, without 1-methylcyclopropene; ±, standard deviation.

**Table 4.** Estimation of the effects of 1-MCP treatment and storage technology on the acidity (% citric acid) of minikiwi after storage.

| Storage Technology | Post-Harvest Treatment | Time of Storage (Weeks) | | | | | | *p*-Value |
|---|---|---|---|---|---|---|---|---|
| | | 2 | 4 | 6 | 8 | 10 | 12 | |
| | | **2015** | | | | | | |
| NA | +1-MCP | $0.93 \pm 0.02$ | $0.96 \pm 0.02$ | $0.90 \pm 0.03$ | $0.86 \pm 0.04$ | - | - | 0.064 |
| | −1-MCP | $0.92 \pm 0.03$ | $0.86 \pm 0.03$ | $0.79 \pm 0.04$ | $0.69 \pm 0.04$ | - | - | <0.001 |
| | *p*-value | 0.714 | 0.026 | 0.031 | 0.010 | | | |
| ULO | +1-MCP | $0.93 \pm 0.03$ | $0.96 \pm 0.04$ | $0.95 \pm 0.06$ | $0.84 \pm 0.02$ | - | - | 0.060 |
| | −1-MCP | $0.88 \pm 0.03$ | $0.90 \pm 0.02$ | $0.83 \pm 0.02$ | $0.80 \pm 0.04$ | - | - | 0.019 |
| | *p*-value | 0.200 | 0.119 | 0.058 | 0.231 | | | |
| | | **2017** | | | | | | |
| NA | +1-MCP | $0.96 \pm 0.05$ | $0.90 \pm 0.05$ | $0.86 \pm 0.10$ | $0.81 \pm 0.03$ | $0.77 \pm 0.06$ | $0.72 \pm 0.01$ | 0.013 |
| | −1-MCP | $0.91 \pm 0.03$ | $0.82 \pm 0.04$ | $0.80 \pm 0.09$ | $0.65 \pm 0.04$ | $0.57 \pm 0.01$ | $0.46 \pm 0.06$ | <0.001 |
| | *p*-value | 0.333 | 0.140 | 0.567 | 0.010 | 0.010 | 0.003 | |
| ULO | +1-MCP | $0.89 \pm 0.07$ | $0.89 \pm 0.02$ | $0.89 \pm 0.10$ | $0.80 \pm 0.06$ | $0.78 \pm 0.06$ | $0.78 \pm 0.04$ | 0.272 |
| | −1-MCP | $0.92 \pm 0.08$ | $0.84 \pm 0.03$ | $0.76 \pm 0.02$ | $0.76 \pm 0.05$ | $0.65 \pm 0.07$ | $0.53 \pm 0.03$ | <0.001 |
| | *p*-value | 0.658 | 0.131 | 0.159 | 0.555 | 0.121 | <0.001 | |

NA, conventional cold storage, 0.1% $CO_2$:21.0% $O_2$; ULO, ultra-low oxygen, 1.5% $CO_2$:1.5% $O_2$; +1-MCP, with 1-methylcyclopropene; −1-MCP, without 1-methylcyclopropene; ±, standard deviation.

Simple sugars are among the basic ingredients that provide energy to the human body, but the polysaccharides contained in *A. arguta* inhibit the oxidation of polyunsaturated fatty acids in food [63]. Sucrose as well as fructose and glucose are found in the highest amounts in the fruits of various Actinidia species. In our study, we found that the glucose content was not determined by the post-harvest application of 1-MCP (Table S2). The storage period appeared to be a significant determinant of the glucose content, as there was a consistent increase in the glucose content over successive analysis dates. Additionally, an interaction of the effect of 1-MCP and the storage period was noted. It turned out that on individual analysis dates, the fruit treated with 1-MCP had a lower glucose content than the untreated fruit (Table 5). Also, technology and, especially, storage at ULO influenced the lower glucose values recorded in the 1-MCP-treated fruit. The interaction of all three factors of the experiment determined the changes in the glucose content of the fruit. It appeared that an upward trend was recorded up to the eighth week of storage in all evaluated combinations of the experiment. In contrast, after 10 and 12 weeks of storage, the glucose content decreased in the untreated fruit, while a further increase in the glucose content was observed in the 1-MCP-treated fruit. The fructose content was determined by 1-MCP (Tables 6 and S2). By evaluating the changes in the fructose content during the experiment, we can conclude that the amount of fructose fluctuated similarly to the glucose content. On most of the analysis dates, it was proven that the fruits treated with 1-MCP were characterised by lower contents of both simple sugars. As in the case of glucose, a steady increase in the fructose content was observed up to week 8, while after weeks 10 and 12, a stabilisation of its content was found in the fruit that was not treated with 1-MCP. In addition, the fruit stored in ULO technology and treated with 1-MCP had the lowest fructose levels.

**Table 5.** Estimation of the effects of 1-MCP treatment and storage technology on the glucose content $(mg \cdot 100 \ g^{-1} \ FW)$ of minikiwi after storage.

| Storage Technology | Post-Harvest Treatment | Time of Storage (Weeks) | | | | | | *p*-Value |
|---|---|---|---|---|---|---|---|---|
| | | **2** | **4** | **6** | **8** | **10** | **12** | |
| | | **2015** | | | | | | |
| NA | +1-MCP | 1.88 ± 0.05 | 2.10 ± 0.03 | 2.38 ± 0.02 | 3.18 ± 0.09 | - | - | <0.001 |
| | −1-MCP | 2.01 ± 0.02 | 2.28 ± 0.07 | 2.64 ± 0.04 | 3.50 ± 0.04 | - | - | <0.001 |
| | *p*-value | 0.026 | 0.027 | <0.001 | 0.010 | | | |
| ULO | +1-MCP | 1.86 ± 0.05 | 2.00 ± 0.02 | 2.31 ± 0.08 | 2.71 ± 0.04 | - | - | <0.001 |
| | −1-MCP | 1.90 ± 0.04 | 2.05 ± 0.04 | 2.51 ± 0.02 | 3.10 ± 0.03 | - | - | <0.001 |
| | *p*-value | 0.446 | 0.162 | 0.025 | <0.001 | | | |
| | | **2017** | | | | | | |
| NA | +1-MCP | 2.11 ± 0.04 | 2.39 ± 0.06 | 2.64 ± 0.04 | 3.12 ± 0.03 | 3.28 ± 0.01 | 3.34 ± 0.04 | <0.001 |
| | −1-MCP | 2.21 ± 0.05 | 2.61 ± 0.11 | 2.98 ± 0.03 | 3.24 ± 0.02 | 3.15 ± 0.01 | 3.02 ± 0.03 | <0.001 |
| | *p*-value | 0.098 | 0.078 | <0.001 | 0.012 | <0.001 | <0.001 | |
| ULO | +1-MCP | 1.96 ± 0.02 | 2.18 ± 0.05 | 2.43 ± 0.06 | 2.77 ± 0.04 | 2.95 ± 0.07 | 3.13 ± 0.07 | <0.001 |
| | −1-MCP | 2.10 ± 0.01 | 2.35 ± 0.06 | 2.73 ± 0.01 | 3.23 ± 0.02 | 3.11 ± 0.02 | 3.00 ± 0.02 | <0.001 |
| | *p*-value | 0.001 | 0.037 | 0.002 | <0.001 | 0.035 | 0.060 | |

NA, conventional cold storage, 0.1% $CO_2$:21.0% $O_2$; ULO, ultra-low oxygen, 1.5% $CO_2$:1.5% $O_2$; +1-MCP, with 1-methylcyclopropene; −1-MCP, without 1-methylcyclopropene; FW, fresh weight; ±, standard deviation.

**Table 6.** Estimation of the effects of 1-MCP treatment and storage technology on the fructose content $(mg \cdot 100 \ g^{-1} \ FW)$ of minikiwi after storage.

| Storage Technology | Post-Harvest Treatment | Time of Storage (Weeks) | | | | | | *p*-Value |
|---|---|---|---|---|---|---|---|---|
| | | **2** | **4** | **6** | **8** | **10** | **12** | |
| | | **2015** | | | | | | |
| NA | +1-MCP | 2.36 ± 0.04 | 2.43 ± 0.07 | 2.64 ± 0.08 | 3.32 ± 0.09 | - | - | <0.001 |
| | −1-MCP | 2.48 ± 0.01 | 2.87 ± 0.04 | 3.18 ± 0.07 | 3.71 ± 0.09 | - | - | <0.001 |
| | *p*-value | 0.010 | 0.001 | 0.001 | 0.012 | | | |
| ULO | +1-MCP | 2.28 ± 0.05 | 2.34 ± 0.04 | 2.50 ± 0.06 | 2.92 ± 0.10 | - | - | <0.001 |
| | −1-MCP | 2.33 ± 0.04 | 2.55 ± 0.03 | 2.70 ± 0.11 | 3.23 ± 0.10 | - | - | <0.001 |
| | *p*-value | 0.316 | 0.005 | 0.072 | 0.039 | | | |
| | | **2017** | | | | | | |
| NA | +1-MCP | 2.72 ± 0.03 | 2.88 ± 0.02 | 3.10 ± 0.05 | 3.59 ± 0.03 | 3.76 ± 0.05 | 3.92 ± 0.06 | <0.001 |
| | −1-MCP | 2.66 ± 0.07 | 3.08 ± 0.07 | 3.46 ± 0.06 | 3.74 ± 0.07 | 3.71 ± 0.05 | 3.67 ± 0.04 | <0.001 |
| | *p*-value | 0.269 | 0.013 | 0.002 | 0.041 | 0.396 | 0.009 | |
| ULO | +1-MCP | 2.59 ± 0.07 | 2.78 ± 0.05 | 2.96 ± 0.06 | 3.25 ± 0.09 | 3.33 ± 0.05 | 3.46 ± 0.05 | <0.001 |
| | −1-MCP | 2.65 ± 0.03 | 2.91 ± 0.10 | 3.14 ± 0.06 | 3.60 ± 0.07 | 3.55 ± 0.01 | 3.53 ± 0.02 | <0.001 |
| | *p*-value | 0.327 | 0.183 | 0.036 | 0.012 | 0.002 | 0.143 | |

NA, conventional cold storage, 0.1% $CO_2$:21.0% $O_2$; ULO, ultra-low oxygen, 1.5% $CO_2$:1.5% $O_2$; +1-MCP, with 1-methylcyclopropene; −1-MCP, without 1-methylcyclopropene; FW, fresh weight; ±, standard deviation.

Different trends were observed for the sucrose content. The disaccharide content was determined by the effect of 1-MCP (Tables 7 and S2). On most evaluation dates, higher sucrose contents were found in the fruit treated with this inhibitor than in the fruit that was not treated with 1-MCP. Changes in the sucrose content occurred in the fruit in the



opposite direction to that of monosaccharides, and sucrose loss was particularly observed in the fruit that was not treated with 1-MCP. An analysis of the data showed that in the second season of storage, sucrose loss in the fruits treated with 1-MCP was not determined by the storage period, which may indicate a higher stability of this disaccharide in these fruits. As in the case of the SSC, the sucrose content was higher in the fruit stored in ULO and treated with 1-MCP than in the other combinations of the experiment. The results obtained in the experiment harmonise with the reports by other authors [64–66]. Wojdyło et al. [65] found a fructose content of 1.16–2.14 g·100 g$^{-1}$ FW, a glucose content of 1.57–2.99 g·100 g$^{-1}$ FW and a sucrose content of 2.41–5.94 g·100 g$^{-1}$ FW. Most reports in the literature report that the contents of monosaccharides in kiwi (*A. deliciosa* and *A. chinensis*) increase during cold storage. Barbonii et al. [67] even found an increase of more than 2-fold in the monosaccharides assessed. There is no information in the literature about the effect of 1-MCP on the content of individual sugars in minikiwi fruit. The content of total sugars and the proportions of these sugars vary not only with the fruit maturity, but also with the cultivar. In a study of kiwifruit (*A. deliciosa* and *A. chinensis*), Nishiyama et al. [66] observed a range of total sugar concentrations from 6150.86 to 10,470.60 g/100 g FW. As can be seen, there is a clear contrast with kiwifruit compared to minikiwi, where the glucose and fructose levels are lower and there is a two- to six-fold predominance of sucrose. The sugar content of the fruit has a significant impact on the management of blood sugar levels after consumption. The glycaemic index (GI) values for the different species of actinidia (*A. deliciosa*, *A. chinensis*, *A. arguta* and *A. rufa*) are relatively low, and low GI values are observed in both healthy people and in people with type II diabetes [68,69].

**Table 7.** Estimation of the effects of 1-MCP treatment and storage technology on the sucrose content (mg·100 g$^{-1}$ FW) of minikiwi after storage.

| Storage Technology | Post-Harvest Treatment | Time of Storage (Weeks) | | | | | | *p*-Value |
|---|---|---|---|---|---|---|---|---|
| | | 2 | 4 | 6 | 8 | 10 | 12 | |
| | | **2015** | | | | | | |
| NA | +1-MCP | 6.52 ± 0.18 | 6.57 ± 0.09 | 6.12 ± 0.17 | 5.84 ± 0.12 | - | - | <0.001 |
| | −1-MCP | 5.87 ± 0.14 | 5.93 ± 0.08 | 5.61 ± 0.10 | 5.25 ± 0.09 | - | - | <0.001 |
| | *p*-value | 0.016 | 0.001 | 0.021 | 0.004 | | | |
| ULO | +1-MCP | 6.75 ± 0.10 | 6.39 ± 0.13 | 6.31 ± 0.18 | 6.18 ± 0.16 | - | - | 0.021 |
| | −1-MCP | 6.49 ± 0.06 | 6.32 ± 0.16 | 5.93 ± 0.04 | 5.72 ± 0.17 | - | - | <0.001 |
| | *p*-value | 0.036 | 0.676 | 0.043 | 0.051 | | | |
| | | **2017** | | | | | | |
| NA | +1-MCP | 5.68 ± 0.37 | 5.69 ± 0.09 | 5.63 ± 0.18 | 5.55 ± 0.08 | 5.38 ± 0.09 | 5.05 ± 0.23 | 0.055 |
| | −1-MCP | 4.91 ± 0.15 | 4.53 ± 0.18 | 4.59 ± 0.04 | 4.53 ± 0.24 | 4.20 ± 0.14 | 3.74 ± 0.22 | <0.001 |
| | *p*-value | 0.051 | 0.001 | 0.001 | 0.004 | <0.001 | 0.004 | |
| ULO | +1-MCP | 5.98 ± 0.23 | 5.81 ± 0.20 | 5.73 ± 0.26 | 5.80 ± 0.13 | 5.67 ± 0.10 | 5.55 ± 0.11 | 0.348 |
| | −1-MCP | 5.39 ± 0.24 | 5.05 ± 0.24 | 5.16 ± 0.06 | 4.90 ± 0.13 | 4.75 ± 0.12 | 4.81 ± 0.24 | 0.039 |
| | *p*-value | 0.066 | 0.026 | 0.041 | 0.002 | 0.001 | 0.016 | |

NA, conventional cold storage, 0.1% $CO_2$:21.0% $O_2$; ULO, ultra-low oxygen, 1.5% $CO_2$:1.5% $O_2$; +1-MCP, with 1-methylcyclopropene; −1-MCP, without 1-methylcyclopropene; FW, fresh weight; ±, standard deviation.

In the ready-to-eat fruit, sugars provide the attractive sweet taste of the kiwifruit, which is balanced by the acid composition. In this study, two primary organic acids, citric and malic acids, were assessed to determine the acidity of minikiwi, with citric acid being several times higher than malic acid (Tables 8, 9 and S2). 1-MCP implied the content of both acids, and the treated fruit were characterised by a higher acid content. As with the previous indicators, the longer storage of minikiwi exacerbated the differences in the contents of both acids between the 1-MCP treated and untreated fruits. This indicates a more rapid loss of acids in the fruit that was not treated with 1-MCP. Although the storage

conditions alone did not have an effect on this indicator, the effect of the inhibitor also interacted with the storage technology. The study confirmed the reports in the literature stating that under ULO conditions, the ripening process is slower [23,25], and the additional inhibition of enzyme activity [33,62] contributes to a synergistic reduction in acid loss by both agents.

**Table 8.** Estimation of the effects of 1-MCP treatment and storage technology on the citric acid content (mg·100 g$^{-1}$ FW) of minikiwi after storage.

| Storage Technology | Post-Harvest Treatment | Time of Storage (Weeks) | | | | | | *p*-Value |
|---|---|---|---|---|---|---|---|---|
| | | **2** | **4** | **6** | **8** | **10** | **12** | |
| | | **2015** | | | | | | |
| NA | +1-MCP | 0.65 ± 0.02 | 0.63 ± 0.00 | 0.61 ± 0.01 | 0.61 ± 0.01 | - | - | 0.034 |
| | −1-MCP | 0.64 ± 0.01 | 0.54 ± 0.01 | 0.53 ± 0.00 | 0.50 ± 0.01 | - | - | <0.001 |
| | *p*-value | 0.759 | <0.001 | <0.001 | <0.001 | | | |
| ULO | +1-MCP | 0.64 ± 0.01 | 0.65 ± 0.02 | 0.59 ± 0.02 | 0.59 ± 0.02 | - | - | 0.017 |
| | −1-MCP | 0.63 ± 0.02 | 0.59 ± 0.01 | 0.56 ± 0.01 | 0.54 ± 0.01 | - | - | 0.001 |
| | *p*-value | 0.611 | 0.008 | 0.111 | 0.030 | | | |
| | | **2017** | | | | | | |
| NA | +1-MCP | 0.81 ± 0.03 | 0.79 ± 0.01 | 0.76 ± 0.06 | 0.70 ± 0.03 | 0.70 ± 0.03 | 0.67 ± 0.01 | 0.007 |
| | −1-MCP | 0.77 ± 0.01 | 0.70 ± 0.01 | 0.68 ± 0.04 | 0.59 ± 0.03 | 0.53 ± 0.01 | 0.47 ± 0.04 | <0.001 |
| | *p*-value | 0.177 | 0.002 | 0.215 | 0.013 | 0.003 | 0.001 | |
| ULO | +1-MCP | 0.77 ± 0.03 | 0.76 ± 0.01 | 0.74 ± 0.05 | 0.71 ± 0.04 | 0.70 ± 0.02 | 0.69 ± 0.01 | 0.083 |
| | −1-MCP | 0.78 ± 0.03 | 0.73 ± 0.01 | 0.68 ± 0.02 | 0.68 ± 0.03 | 0.62 ± 0.04 | 0.53 ± 0.01 | <0.001 |
| | *p*-value | 0.844 | 0.063 | 0.158 | 0.356 | 0.067 | <0.001 | |

NA, conventional cold storage, 0.1% $CO_2$:21.0% $O_2$; ULO, ultra-low oxygen, 1.5% $CO_2$:1.5% $O_2$; +1-MCP, with 1-methylcyclopropene; −1-MCP, without 1-methylcyclopropene; FW, fresh weight; ±, standard deviation.

**Table 9.** Estimation of the effects of 1-MCP treatment and storage technology on the malic acid content (mg·100 g$^{-1}$ FW) of minikiwi after storage.

| Storage Technology | Post-Harvest Treatment | Time of Storage (Weeks) | | | | | | *p*-Value |
|---|---|---|---|---|---|---|---|---|
| | | **2** | **4** | **6** | **8** | **10** | **12** | |
| | | **2015** | | | | | | |
| NA | +1-MCP | 0.150 ± 0.00 | 0.146 ± 0.01 | 0.143 ± 0.01 | 0.137 ± 0.01 | - | - | 0.002 |
| | −1-MCP | 0.151 ± 0.01 | 0.136 ± 0.01 | 0.116 ± 0.01 | 0.095 ± 0.01 | - | - | <0.001 |
| | *p*-value | 0.576 | 0.018 | <0.001 | <0.001 | | | |
| ULO | +1-MCP | 0.159 ± 0.01 | 0.149 ± 0.01 | 0.145 ± 0.01 | 0.136 ± 0.01 | - | - | 0.001 |
| | −1-MCP | 0.154 ± 0.00 | 0.166 ± 0.01 | 0.139 ± 0.01 | 0.108 ± 0.01 | - | - | <0.001 |
| | *p*-value | 0.177 | <0.001 | 0.162 | <0.001 | | | |
| | | **2017** | | | | | | |
| NA | +1-MCP | 0.124 ± 0.01 | 0.122 ± 0.01 | 0.128 ± 0.01 | 0.103 ± 0.01 | 0.103 ± 0.01 | 0.100 ± 0.04 | <0.001 |
| | −1-MCP | 0.119 ± 0.01 | 0.103 ± 0.01 | 0.094 ± 0.01 | 0.088 ± 0.01 | 0.071 ± 0.01 | 0.057 ± 0.04 | <0.001 |
| | *p*-value | 0.236 | 0.037 | <0.001 | 0.005 | 0.017 | <0.001 | |
| ULO | +1-MCP | 0.125 ± 0.01 | 0.117 ± 0.01 | 0.114 ± 0.01 | 0.105 ± 0.01 | 0.106 ± 0.01 | 0.103 ± 0.01 | 0.011 |
| | −1-MCP | 0.124 ± 0.01 | 0.117 ± 0.01 | 0.109 ± 0.01 | 0.096 ± 0.01 | 0.089 ± 0.01 | 0.076 ± 0.01 | <0.001 |
| | *p*-value | 0.901 | 0.978 | 0.260 | 0.062 | 0.016 | 0.018 | |

NA, conventional cold storage, 0.1% $CO_2$:21.0% $O_2$; ULO, ultra-low oxygen, 1.5% $CO_2$:1.5% $O_2$; +1-MCP, with 1-methylcyclopropene; −1-MCP, without 1-methylcyclopropene; FW, fresh weight; ±, standard deviation.

*3.3. Antioxidant Properties*

The antioxidant properties of minikiwi are related to the high content of ascorbic acid [7,16,18]. Changes in the content of antioxidant compounds and their direction occurring during storage are determined by the fruit maturity [22,41], the conditions under which the fruit was stored [20,23] and the post-harvest treatment [39]. Most authors report a reduction in the ascorbic acid content during storage [45,70]. The application of 1-MCP definitely delayed the decrease in the ascorbic acid content during storage (Tables 10 and S3). The positive effect of 1-MCP was particularly evident in the second season of the study. The inhibitory effect of 1-MCP is confirmed by the absence of significant changes in the ascorbic acid content in the fruit treated with 1-MCP as well as an increase in the disparity in the acid content between the untreated fruit and the fruit treated with 1-MCP during storage. Also, Lim et al. [39] report that the application of 1-MCP effectively inhibits the loss of ascorbic acid during conventional cold storage. The application of 1-MCP may contribute to maintaining the high antioxidant properties of minikiwi during the fruit distribution period. The experiment did not show any effect of the storage technology on the ascorbic acid content, and the effect of the storage time was only detected in the second year of the study. The second season also showed interactions between 1-MCP and the storage technology and between 1-MCP and the storage time, which may be related to the extension of the storage period in 2017. Notably, the application of the inhibitor also had the effect of inhibiting the loss of total polyphenols (TPC). Again, the results of the study indicate that the application of 1-MCP, irrespective of the conditions under which the minikiwi was stored, works well in reducing TPC loss. In both storage seasons, significantly higher TPC values were recorded for the 1-MCP-treated fruit than for the untreated fruit, and even extending the storage period to 12 weeks exacerbated the differences in the TPC content between the treated and untreated fruits. The experiment showed a positive interaction between 1-MCP and the storage technology, but only in the first storage season. The use of ULO in combination with 1-MCP proved to be the most effective solution to maintain a high total polyphenol content. Many authors report that the use of advanced technologies such as ULO or a controlled atmosphere, by lowering the $O_2$ and raising the $CO_2$, inhibits TPC loss in various fruit species [19,23,29]. Also, the results of our own research indicate that the use of ULO technology is significantly more efficient than conventional cold storage (NA). However, the information in the literature is inconclusive, as few researchers have observed an increase in the TPC content during minikiwi storage [19]. Previous information in the literature suggests that the use of 1-MCP is highly effective in maintaining high fruit potential [5,39], which is supported by the results of our own study. The inhibitory character of 1-MCP on the derivatives of phenolic acids, flavonols and flavan-3-ols was demonstrated in the experiment, but the obtained results were slightly modified in the different storage seasons (Tables 11–14 and S3). The strongest effect of 1-MCP was found for flavonol and flavan-3-ol derivatives, as this effect was proven in both years of the study. For phenolic acid, the analysis showed a significant effect in only one of the two storage seasons. Nevertheless, in each case, the effect of 1-MCP on the fruit was in one direction—it inhibited the loss of compounds from the different phenolic groups compared to the fruit that was not treated with 1-MCP. As mentioned, the influence of the experimental factors depended on the year of the study. In the first storage season, it appeared that only the storage technology determined the changes in the phenolic acid derivatives, while in the following season, the effects of 1-MCP and the storage period was shown. In the case of flavonols, only 1-MCP was a factor that determined the changes in their content in both years, while the flavan-3-ols were affected by the influences of 1-MCP and the storage period, without the contribution of the storage technology. The variable influence of the individual factors observed in the study years is often related to other causes, such as the weather conditions of a given year [8,51,63] or the state of fruit maturity [22,41]. Regardless of the year of study, 1-MCP was the main factor limiting the loss of compounds from the polyphenol group, as was the case with the flavan-3-ol content,

which was not reduced in the fruit treated with 1-MCP. Many authors signal that fruits after 1-MCP treatment show high values of compounds from the phenolic group [5,29,31,39].

**Table 10.** Estimation of the effects of 1-MCP treatment and storage technology on the ascorbic acid content (mg·100 g$^{-1}$ FW) of minikiwi after storage.

| Storage Technology | Post-Harvest Treatment | Time of Storage (Weeks) | | | | | | *p*-Value |
|---|---|---|---|---|---|---|---|---|
| | | 2 | 4 | 6 | 8 | 10 | 12 | |
| | | **2015** | | | | | | |
| NA | +1-MCP | 87.9 ± 7.0 | 81.0 ± 4.3 | 84.4 ± 9.0 | 78.0 ± 5.4 | - | - | 0.515 |
| | −1-MCP | 76.2 ± 5.1 | 67.2 ± 4.1 | 67.0 ± 6.3 | 63.2 ± 1.6 | - | - | 0.094 |
| | *p*-value | 0.136 | 0.031 | 0.088 | 0.020 | | | |
| ULO | +1-MCP | 84.0 ± 5.8 | 82.6 ± 3.3 | 83.2 ± 4.8 | 80.0 ± 4.1 | - | - | 0.832 |
| | −1-MCP | 78.8 ± 1.5 | 73.7 ± 4.6 | 70.6 ± 2.5 | 67.8 ± 3.7 | - | | 0.049 |
| | *p*-value | 0.284 | 0.090 | 0.030 | 0.034 | | | |
| | | **2017** | | | | | | |
| NA | +1-MCP | 79.7 ± 3.9 | 75.8 ± 2.6 | 70.6 ± 4.2 | 75.5 ± 4.7 | 72.9 ± 0.2 | 72.0 ± 0.3 | 0.133 |
| | −1-MCP | 67.4 ± 2.1 | 66.3 ± 2.7 | 60.4 ± 1.2 | 60.2 ± 3.6 | 54.7 ± 1.0 | 50.9 ± 1.0 | <0.001 |
| | *p*-value | 0.017 | 0.023 | 0.028 | 0.021 | <0.001 | <0.001 | |
| ULO | +1-MCP | 76.6 ± 3.8 | 74.0 ± 2.8 | 72.4 ± 1.8 | 72.9 ± 1.8 | 70.9 ± 0.3 | 69.7 ± 0.3 | 0.101 |
| | −1-MCP | 71.4 ± 2.1 | 65.2 ± 3.2 | 65.0 ± 2.0 | 66.1 ± 0.9 | 59.5 ± 0.8 | 56.4 ± 0.8 | <0.001 |
| | *p*-value | 0.166 | 0.043 | 0.016 | 0.008 | <0.001 | <0.001 | |

NA, conventional cold storage, 0.1% $CO_2$:21.0% $O_2$; ULO, ultra-low oxygen, 1.5% $CO_2$:1.5% $O_2$; +1-MCP, with 1-methylcyclopropene; −1-MCP, without 1-methylcyclopropene; FW, fresh weight; ±, standard deviation.

**Table 11.** Estimation of the effects of 1-MCP treatment and storage technology on the total polyphenol content (mg·100 g$^{-1}$ FW) of minikiwi after storage.

| Storage Technology | Post-Harvest Treatment | Time of Storage (Weeks) | | | | | | *p*-Value |
|---|---|---|---|---|---|---|---|---|
| | | 2 | 4 | 6 | 8 | 10 | 12 | |
| | | **2015** | | | | | | |
| NA | +1-MCP | 97.7 ± 3.4 | 94.1 ± 3.8 | 94.2 ± 2.4 | 90.7 ± 3.0 | - | - | 0.259 |
| | −1-MCP | 91.2 ± 2.9 | 84.5 ± 2.4 | 81.3 ± 3.5 | 78.2 ± 1.9 | - | - | 0.008 |
| | *p*-value | 0.108 | 0.039 | 0.012 | 0.007 | | | |
| ULO | +1-MCP | 102.2 ± 6.8 | 100.1 ± 2.2 | 101.7 ± 2.5 | 101.9 ± 2.3 | - | - | 0.949 |
| | −1-MCP | 92.2 ± 4.9 | 81.6 ± 6.1 | 77.8 ± 6.8 | 75.7 ± 3.0 | - | - | 0.060 |
| | *p*-value | 0.165 | 0.015 | 0.009 | <0.001 | | | |
| | | **2017** | | | | | | |
| NA | +1-MCP | 113.5 ± 3.0 | 108.2 ± 6.0 | 105.6 ± 7.2 | 107.7 ± 5.4 | 100.5 ± 0.1 | 99.8 ± 0.3 | 0.085 |
| | −1-MCP | 95.6 ± 2.4 | 88.7 ± 2.1 | 84.5 ± 7.1 | 95.3 ± 2.3 | 74.4 ± 1.3 | 69.1 ± 1.0 | <0.001 |
| | *p*-value | 0.002 | 0.012 | 0.014 | 0.040 | <0.001 | <0.001 | |
| ULO | +1-MCP | 111.8 ± 7.6 | 111.9 ± 3.8 | 106.7 ± 5.4 | 106.8 ± 5.7 | 103.3 ± 0.5 | 104.9 ± 2.3 | 0.414 |
| | −1-MCP | 97.0 ± 7.3 | 87.7 ± 4.0 | 80.8 ± 2.3 | 83.4 ± 4.1 | 84.1 ± 3.6 | 79.9 ± 1.6 | 0.016 |
| | *p*-value | 0.117 | 0.003 | 0.003 | 0.008 | 0.001 | <0.001 | |

NA, conventional cold storage, 0.1% $CO_2$:21.0% $O_2$; ULO, ultra-low oxygen, 1.5% $CO_2$:1.5% $O_2$; +1-MCP, with 1-methylcyclopropene; −1-MCP, without 1-methylcyclopropene; FW, fresh weight; ±, standard deviation.

**Table 12.** Estimation of the effects of 1-MCP treatment and storage technology on the phenolic acid content (mg·100 g$^{-1}$ FW) of minikiwi after storage.

| Storage Technology | Post-Harvest Treatment | Time of Storage (Weeks) | | | | | | *p*-Value |
|---|---|---|---|---|---|---|---|---|
| | | **2** | **4** | **6** | **8** | **10** | **12** | |
| | | **2015** | | | | | | |
| NA | +1-MCP | $1.66 \pm 0.13$ | $1.57 \pm 0.27$ | $1.52 \pm 0.31$ | $1.49 \pm 0.29$ | - | - | 0.916 |
| | −1-MCP | $1.61 \pm 0.05$ | $1.55 \pm 0.09$ | $1.38 \pm 0.10$ | $1.36 \pm 0.19$ | | | 0.170 |
| | *p*-value | 0.630 | 0.951 | 0.574 | 0.611 | | | |
| ULO | +1-MCP | $1.68 \pm 0.13$ | $1.73 \pm 0.10$ | $1.66 \pm 0.28$ | $1.67 \pm 0.09$ | - | - | 0.974 |
| | −1-MCP | $1.74 \pm 0.05$ | $1.69 \pm 0.11$ | $1.54 \pm 0.19$ | $1.53 \pm 0.16$ | - | - | 0.377 |
| | *p*-value | 0.662 | 0.725 | 0.638 | 0.317 | | | |
| | | **2017** | | | | | | |
| NA | +1-MCP | $2.21 \pm 0.05$ | $2.10 \pm 0.08$ | $2.15 \pm 0.05$ | $2.17 \pm 0.12$ | $2.14 \pm 0.04$ | $2.04 \pm 0.02$ | 0.270 |
| | −1-MCP | $1.98 \pm 0.08$ | $1.81 \pm 0.10$ | $1.54 \pm 0.09$ | $1.65 \pm 0.04$ | $1.56 \pm 0.07$ | $1.51 \pm 0.02$ | <0.001 |
| | *p*-value | 0.026 | 0.033 | <0.001 | 0.005 | <0.001 | <0.001 | |
| ULO | +1-MCP | $2.30 \pm 0.07$ | $2.22 \pm 0.03$ | $2.16 \pm 0.08$ | $2.02 \pm 0.09$ | $2.05 \pm 0.02$ | $2.11 \pm 0.06$ | 0.010 |
| | −1-MCP | $2.08 \pm 0.02$ | $1.91 \pm 0.11$ | $1.75 \pm 0.06$ | $1.79 \pm 0.09$ | $1.59 \pm 0.04$ | $1.42 \pm 0.03$ | <0.001 |
| | *p*-value | 0.016 | 0.021 | 0.005 | 0.067 | <0.001 | <0.001 | |

NA, conventional cold storage, 0.1% $CO_2$:21.0% $O_2$; ULO, ultra-low oxygen, 1.5% $CO_2$:1.5% $O_2$; +1-MCP, with 1-methylcyclopropene; −1-MCP, without 1-methylcyclopropene; FW, fresh weight; ±, standard deviation.

**Table 13.** Estimation of the effects of 1-MCP treatment and storage technology on the flavonol content (mg·100 g$^{-1}$ FW) of minikiwi after storage.

| Storage Technology | Post-Harvest Treatment | Time of Storage (Weeks) | | | | | | *p*-Value |
|---|---|---|---|---|---|---|---|---|
| | | **2** | **4** | **6** | **8** | **10** | **12** | |
| | | **2015** | | | | | | |
| NA | +1-MCP | $5.36 \pm 0.48$ | $5.38 \pm 0.35$ | $5.23 \pm 0.16$ | $5.45 \pm 0.30$ | - | - | 0.930 |
| | −1-MCP | $4.75 \pm 0.28$ | $4.61 \pm 0.07$ | $4.01 \pm 0.21$ | $3.91 \pm 0.19$ | | | 0.006 |
| | *p*-value | 0.188 | 0.036 | 0.003 | 0.003 | | | |
| ULO | +1-MCP | $5.52 \pm 0.15$ | $5.50 \pm 0.24$ | $5.34 \pm 0.19$ | $5.38 \pm 0.31$ | - | - | 0.826 |
| | −1-MCP | $5.10 \pm 0.61$ | $4.42 \pm 0.32$ | $4.08 \pm 0.23$ | $4.18 \pm 0.22$ | - | - | 0.098 |
| | *p*-value | 0.390 | 0.018 | 0.003 | 0.010 | | | |
| | | **2017** | | | | | | |
| NA | +1-MCP | $4.64 \pm 0.19$ | $4.13 \pm 0.22$ | $4.06 \pm 0.28$ | $4.24 \pm 0.23$ | $4.08 \pm 0.07$ | $3.99 \pm 0.10$ | 0.060 |
| | −1-MCP | $3.90 \pm 0.17$ | $3.49 \pm 0.26$ | $3.14 \pm 0.13$ | $3.35 \pm 0.12$ | $3.15 \pm 0.06$ | $2.94 \pm 0.02$ | <0.001 |
| | *p*-value | 0.015 | 0.056 | 0.014 | 0.008 | <0.001 | <0.001 | |
| ULO | +1-MCP | $4.44 \pm 0.28$ | $4.47 \pm 0.28$ | $4.34 \pm 0.21$ | $4.35 \pm 0.08$ | $4.19 \pm 0.06$ | $4.30 \pm 0.06$ | 0.754 |
| | −1-MCP | $3.83 \pm 0.12$ | $3.69 \pm 0.11$ | $3.34 \pm 0.21$ | $3.88 \pm 0.19$ | $3.37 \pm 0.07$ | $3.29 \pm 0.05$ | 0.002 |
| | *p*-value | 0.046 | 0.022 | 0.009 | 0.030 | 0.001 | <0.001 | |

NA, conventional cold storage, 0.1% $CO_2$:21.0% $O_2$; ULO, ultra-low oxygen, 1.5% $CO_2$:1.5% $O_2$; +1-MCP, with 1-methylcyclopropene; −1-MCP, without 1-methylcyclopropene; FW, fresh weight; ±, standard deviation.

**Table 14.** Estimation of the effects of 1-MCP treatment and storage technology on the flavan-3-ol content (mg·100 g$^{-1}$ FW) of minikiwi after storage.

| Storage Technology | Post-Harvest Treatment | Time of Storage (Weeks) | | | | | | *p*-Value |
|---|---|---|---|---|---|---|---|---|
| | | 2 | 4 | 6 | 8 | 10 | 12 | |
| | | **2015** | | | | | | |
| NA | +1-MCP | 0.61 ± 0.02 | 0.62 ± 0.04 | 0.60 ± 0.02 | 0.56 ± 0.03 | - | - | 0.210 |
| | −1-MCP | 0.51 ± 0.05 | 0.47 ± 0.02 | 0.42 ± 0.02 | 0.38 ± 0.03 | - | - | 0.011 |
| | *p*-value | 0.054 | 0.005 | 0.001 | 0.002 | | | |
| ULO | +1-MCP | 0.61 ± 0.04 | 0.62 ± 0.02 | 0.61 ± 0.03 | 0.62 ± 0.04 | - | - | 0.983 |
| | −1-MCP | 0.53 ± 0.02 | 0.51 ± 0.04 | 0.45 ± 0.02 | 0.46 ± 0.10 | - | - | 0.479 |
| | *p*-value | 0.059 | 0.017 | 0.004 | 0.105 | | | |
| | | **2017** | | | | | | |
| NA | +1-MCP | 0.57 ± 0.02 | 0.51 ± 0.03 | 0.52 ± 0.04 | 0.52 ± 0.03 | 0.48 ± 0.01 | 0.50 ± 0.01 | 0.063 |
| | −1-MCP | 0.47 ± 0.03 | 0.45 ± 0.01 | 0.41 ± 0.02 | 0.42 ± 0.05 | 0.41 ± 0.01 | 0.39 ± 0.02 | 0.169 |
| | *p*-value | 0.020 | 0.027 | 0.017 | 0.079 | 0.006 | 0.004 | |
| ULO | +1-MCP | 0.54 ± 0.02 | 0.55 ± 0.01 | 0.51 ± 0.02 | 0.53 ± 0.05 | 0.52 ± 0.02 | 0.52 ± 0.02 | 0.701 |
| | −1-MCP | 0.48 ± 0.02 | 0.45 ± 0.03 | 0.42 ± 0.04 | 0.48 ± 0.02 | 0.39 ± 0.01 | 0.37 ± 0.01 | 0.001 |
| | *p*-value | 0.047 | 0.009 | 0.051 | 0.205 | <0.001 | <0.001 | |

NA, conventional cold storage, 0.1% $CO_2$:21.0% $O_2$; ULO, ultra-low oxygen, 1.5% $CO_2$:1.5% $O_2$; +1-MCP, with 1-methylcyclopropene; −1-MCP, without 1-methylcyclopropene; FW, fresh weight; ±, standard deviation.

The application of 1-MCP inhibited fruit ripening and reduced the loss of ascorbic acid and other compounds from the phenolic group, which influenced the antioxidant activity of minikiwi. The analysis of the test results shows a significant dependence of antioxidant activity on the application of 1-MCP (Table S3). However, the results from the individual analyses only show the significance of this relationship on a few assessment dates (Table 15). This may be related to the slight decrease in AA during the first season studies. Extending the storage period to 12 weeks allowed for the efficacy of 1-MCP to be demonstrated, but only during the additional storage period. The antioxidant activity is determined by the different compounds that are present in the fruit. In minikiwi, ascorbic acid has a high effect on AA [16], but the phenolic content also determines, although to a lesser extent, the antioxidant potential of the fruit [54]. The results obtained are partly consistent with the previous reports on the effect of 1-MCP on the AA of minikiwi [70]. There is a view that AA is reduced during minikiwi ripening under refrigerated conditions, but the rate of this process is dependent on many factors, i.e., the storage temperature [16,19,22], the storage technology [20,23] as well as the use of pre- and post-harvest fruit ripening inhibitors [29,39,56].

**Table 15.** Estimation of the effects of 1-MCP treatment and storage technology on the antioxidant activity (mg·1 g$^{-1}$ FW) of minikiwi after storage.

| Storage Technology | Post-Harvest Treatment | Time of Storage (Weeks) | | | | | | *p*-Value |
|---|---|---|---|---|---|---|---|---|
| | | 2 | 4 | 6 | 8 | 10 | 12 | |
| | | **2015** | | | | | | |
| NA | +1-MCP | 1.02 ± 0.10 | 0.96 ± 0.05 | 0.93 ± 0.08 | 0.97 ± 0.14 | - | - | 0.814 |
| | −1-MCP | 0.86 ± 0.09 | 0.78 ± 0.02 | 0.78 ± 0.04 | 0.79 ± 0.11 | - | - | 0.676 |
| | *p*-value | 0.154 | 0.007 | 0.075 | 0.234 | | | |
| ULO | +1-MCP | 0.99 ± 0.05 | 0.98 ± 0.03 | 0.95 ± 0.05 | 0.98 ± 0.10 | - | - | 0.897 |
| | −1-MCP | 0.91 ± 0.07 | 0.82 ± 0.06 | 0.84 ± 0.03 | 0.81 ± 0.01 | - | - | 0.249 |
| | *p*-value | 0.289 | 0.028 | 0.074 | 0.071 | | | |
| | | **2017** | | | | | | |
| NA | +1-MCP | 0.83 ± 0.02 | 0.82 ± 0.04 | 0.71 ± 0.02 | 0.79 ± 0.05 | 0.78 ± 0.02 | 0.80 ± 0.04 | 0.040 |
| | −1-MCP | 0.73 ± 0.01 | 0.74 ± 0.05 | 0.67 ± 0.02 | 0.69 ± 0.04 | 0.64 ± 0.04 | 0.59 ± 0.02 | 0.004 |
| | *p*-value | 0.003 | 0.155 | 0.123 | 0.090 | 0.012 | 0.001 | |
| ULO | +1-MCP | 0.79 ± 0.01 | 0.81 ± 0.02 | 0.79 ± 0.04 | 0.78 ± 0.06 | 0.76 ± 0.02 | 0.77 ± 0.04 | 0.810 |
| | −1-MCP | 0.78 ± 0.02 | 0.72 ± 0.06 | 0.76 ± 0.02 | 0.71 ± 0.02 | 0.66 ± 0.02 | 0.64 ± 0.04 | 0.009 |
| | *p*-value | 0.974 | 0.111 | 0.300 | 0.202 | 0.011 | 0.033 | |

NA, conventional cold storage, 0.1% $CO_2$:21.0% $O_2$; ULO, ultra-low oxygen, 1.5% $CO_2$:1.5% $O_2$; +1-MCP, with 1-methylcyclopropene; −1-MCP, without 1-methylcyclopropene; FW, fresh weight; ±, standard deviation.

## 4. Conclusions

The use of 1-MCP in fruit storage has been the research focus of recent years, and the results so far are promising. The loss of firmness is the main cause limiting the distribution time of minikiwi fruit. The application of 1-MCP effectively reduces the loss of fruit firmness during storage. The results indicate that by applying this inhibitor after fruit harvest, we can extend the fruit distribution period up to 12 weeks and perhaps longer. The minikiwi fruits treated with 1-MCP were still characterised by a high firmness even after 12 weeks of storage. It should be noted that blocking the ripening of minikiwi by 1-MCP leads, on the one hand, to the preservation of fruit firmness, but the fruit is less sweet and more sour, which may be less acceptable to consumers. In order to obtain higher flavour parameters of minikiwi treated with 1-MCP, it would probably be necessary to extend the storage period. This is probably possible, since minikiwi treated with 1-MCP even after 12 weeks of storage did not reach maximum sweetening, as evidenced by the steady increase in simple sugars. Minikiwi, like most fruits of the actinidia genus, is considered a fruit with a high antioxidant potential. Minikiwi and other fruits lose their potential during storage. 1-Methylcyclopropene proved to be a fairly effective inhibitor of this process, although its effectiveness depended on the season as well as the length of storage. The influence of the year is a strong determinant of the antioxidant properties of fruit, as highlighted by many researchers. Proper fruit handling methodology should prevent a drastic loss of phenolic compounds and vitamin C. The results show that the decomposition inhibitory effect of 1-MCP is stronger for flavan-3-ols, while it is slightly weaker for phenolic acids. However, fruit treated with 1-MCP has a higher antioxidant potential than untreated fruit, and the differences increase with the length of storage. An alternative to 1-MCP is modern storage technologies, but the costliness of such technologies is too high to use them widely for minikiwi storage. The use of 1-MCP is equally effective in blocking fruit ripening while keeping the antioxidant potential of the fruit high. Expanding knowledge and developing a proper procedure for the use of 1-methylcyclopropene in minikiwi storage could enable the fruit to be distributed over a longer period and contribute to its wider use by both consumers and the processing industry.

**Supplementary Materials:** The following supporting information can be downloaded at https://www.mdpi.com/article/10.3390/agriculture13091665/s1, Table S1: The *p*-value of the influence of individual experimental factors and interactions on the physicochemical properties of fruits; Table S2: The *p*-value of the influence of individual experimental factors and interactions on the content of mono and disaccharides in fruits; Table S3: The *p*-value of the influence of individual experimental factors and interactions on antioxidant properties of minikiwi.

**Author Contributions:** Conceptualisation, T.K.; methodology, T.K.; software, T.K. and A.K.; formal analysis, T.K. and A.K.; investigation, T.K. and A.K.; data curation, T.K.; writing—original draft preparation, T.K. and K.T.; writing—review and editing, T.K. and K.T. All authors have read and agreed to the published version of the manuscript.

**Funding:** This research received no external funding.

**Institutional Review Board Statement:** Not applicable.

**Data Availability Statement:** Not applicable.

**Conflicts of Interest:** The authors declare no conflict of interest.

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
