# Peer review of "Estimating the Physicochemical and Antioxidant Properties of Hardy Kiwi (Actinidia arguta) Treated with 1-Methylcyclopropene during Storage"

_agriculture, doi:10.3390/agriculture13091665_

Round 1

Reviewer 1 Report

 1-MCP is an effective fruit ripening inhibitor, in particular for climacteric fruits, including kiwi fruits. The use of 1-MCP in combination with other storage technology may has synergistic effect on the fruit storage. In the manuscript “Estimating the physicochemical and antioxidant properties of 2 hardy kiwi (Actinidia arguta) treated with 1-MCP during storage”, Krupa et al. first characterized the physicochemical and antioxidant properties of minikiwi (Actinidia arguta), and found some distinct properties from other kiwi fruits. They also found that 1-MCP combined with normal cold storage (NA) or under low oxygen concentration (ULO) had a synergistic effect on the storage, with detailed study on the effect of the antioxidant activity. The experiments were well designed and the findings regarding the effect of 1-MCP on the antioxidant activity is interesting. The manuscript is written clearly and worth publishing. However, some concerns are the following:

1.       In the notes of the tables, the units all for the indicators are lack.

2.       There should be various flavonols and flavan-3ols in fruits, the separation profiles and identification of the phenolic compounds with standards by HPLC are required in  supplementary data to show the accuracy of the determination.

Minor concerns

Some writing of the data is not followed the format e. g. the upper case of “-1” in “mg·100 g-1 FW “, and The lower case of 2 in O2 etc..

Reviewer 2 Report

1. The conclusion part of the abstract is too small, only two sentences.

2. What do +1-MCP and -1-MCP represent? In the current version of the manuscript, I have difficulty finding the actual meaning of what they represent.

3. The current form of table presentation is not conducive to understanding the experimental results. It is recommended that the author consider converting it into a form of chart display.

4. Appearance is also the main component of fruit quality. If the author provides pictures of the appearance of minikiwi during storage, it will be more effective to support the effectiveness of the treatment.

5. The authors monitored CO2 and O2 levels, so why not provide data on respiration during minikiwi storage? This is a key indicator of the physiological properties of minikiwi during storage.

none

Round 2

Reviewer 2 Report

none